# qlifetable: An R package for constructing quarterly life tables

**Jose M. Pavía** *, **Josep Lledó**

Universitat de Valencia, Valencia, Spain

* pavia@uv.es

**Data Availability Statement:** The data handled in this paper is the data are held in a public repositories: <https://data.mendeley.com/datasets/39g9fgxg56/1>, <https://data.mendeley.com/datasets/3wv5wjk9tc/1> and <https://CRAN.R-project.org/package=qlifetable>.

## Abstract

The big data revolution has greatly expanded the availability of microdata on vital statistics, providing researchers with unprecedented access to large and complex datasets on birth, death, migration, and population, sometimes even including exact dates of demographic events. This has led to the development of a novel methodology for estimating sub-annual life tables that offers new opportunities for the insurance industry, also potentially impacting on the management of pension funds and social security systems. This paper introduces the qlifetable package, an R implementation of this methodology. It begins by detailing how basic summary statistics are computed by the package from detailed individual records, including the length of age years, which should be observed as relative (subjective) to ensure congruency between age and calendar time when measuring exposure times and exact ages of individuals at events. This is a new result that compels the observation of time as relative in the disciplines of actuarial science, risk management and demography. Afterwards, the paper demonstrates the use of the package, which integrates a set of functions for estimating crude quarterly (and annual) death rates, calculating seasonal-ageing indexes (SAIs) and building quarterly life tables for a (general or insured) population by exploiting either microdata of dates of births and events or summary statistics.

## 1. Introduction

A life table is a statistical tool that synthesizes the mortality patterns of a population. Annual life tables are routinely employed by many agents, including demographers, policymakers, and actuaries of insurance companies. Demographers are primary users of life tables. Among other uses, they employ life tables to analyse population dynamics, life expectancies and mortality and survival rates, including the comparison of population trends and mortality rates across different regions or subpopulations. Life tables are also a key resource for social planners and policymakers, who use them to develop long-term projections and plans that impact the social and economic dimensions of nations [1].

Life tables are even more critical for public social security, the insurance industry and to manage pension funds. On the one hand, they inform future demand for retirement benefits and the financial sustainability of social security systems. On the other hand, they are essential for calculating premiums, benefits, and retirement pay-outs. Life tables constitute the fulcrum

**Funding:** This work was supported by Generalitat Valenciana, Conselleria de Educación, Cultura, Universidades y Empleo under Grants CIAICO/2023/031 and CIGE/2023/7 awarded to JMP and JL, respectively; Ministerio de Ciencia e Innovación under Grant PID2021-128228NB-I00 awarded to both JMP and JL; and Fundación Mapfre under Grant "Modelización espacial e intra-anual de la mortalidad en España. Una herramienta automática para el cálculo de productos de vida" awarded to both JMP and JL. The funders had no role in study design, data collection and analysis, decision to publish, or preparation of the manuscript.

**Competing interests:** The authors have declared that no competing interests exist.

on which the life insurance business rests [2]. Other uses of life tables can be found in market research and the pharmaceutical industry, where they are used to segment populations by age, sex and other factors, and for developing products that address the needs of specific age groups.

Overall, life tables and death rates are key instruments spanning many disciplines and with multiple applications. It is not surprising, therefore, that over recent years much attention has been given to evolving life expectancy and mortality rate forecasting models, including stochastic, functional data and Bayesian models [3–5], and to improving the estimators of death probabilities [6–8].

The big data revolution has greatly expanded the availability of microdata on vital statistics, providing researchers with unprecedented access to large and complex datasets on birth, death, migration flows, and population, sometimes even including exact dates of occurrence of demographic events. This has opened up new opportunities for exploring the underlying causes and drivers of mortality trends and for improving general population death rate estimates. Indeed, big microdata (including the millions of data corresponding to population censuses) are already being incorporated into the construction of annual life tables [9, 10] and have very recently paved the way for building quarterly life tables that take into account the quarter of the year in which the birth takes place [11]. Certainly, the arrival of the big data and information technology revolution has fueled the development of a new methodology to estimate sub-annual life tables in both time dimensions: age and calendar.

The issues with the new approach arise because fully exploit its potential typically involves processing millions of microdata records—although, as we will show, approximations can be obtained by processing just a few thousand data records—and because defining/calculating some of the required summary statistics can be problematic due to leap years. This paper introduces the R-package qlifetable, an R implementation of this methodology accessible on CRAN, that automatizes the process of estimating all the required statistics, including seasonal-ageing indexes (SAIs) for constructing quarterly life tables.

The aim of this paper is twofold. On the one hand, it details how basic computations are performed within the package to summarize microdata records, particularly regarding the measurement of exposure times. While this task poses no special difficulties when considering a single temporal dimension or working on an annual basis, it becomes notably challenging when addressing two temporal dimensions and dealing with sub-annual periods. On the other hand, the paper exemplifies the use of the different functions included in qlifetable and describes the complete process for producing quarterly life tables.

The rest of the paper is structured as follows. Section 2 shows, with the help of the Lexis diagram, how the quarterly summary statistics are computed. Section 3 describes how to directly estimate quarterly life tables and how to derive them from annual life tables using seasonal-ageing indexes (SAIs). Section 4 explains how to accurately measure ages at events while accounting for the disruptive impact of leap years and discusses the use of person-dependent age years to address this issue. Section 5 presents the package and summarises its main functions, with Section 6 illustrating how to use them. The final section concludes.

## 2. Basic statistics. The Lexis scheme

Period annual and quarterly life tables are constructed using (central) mortality rates as seeds. Both crude annual and quarterly mortality rates are estimated from observed demographic events as the ratio of deaths in the population per unit of time-exposure. The annual mortality rate at age $x$ during annum $a$, $m_x^a$, is estimated as the quotient between the number of deaths recorded in the population with age $x$ last birthday during year $a$ and the total time exposed to

the risk of death with age $x$ by the population during year $a$. Similarly, a raw estimate of the quarterly mortality rate at a specific age $x$ during a particular age-quarter $r$ and calendar-quarter $s$ of year $a$, ${}_s^r m_x^a$, is obtained by dividing the number of deaths recorded in the population with age $x$ during that particular quarter of year $a$ by the total time that all the members of the population were at risk of death (i.e., exposed to mortality risk) during the $(r, s)$-quarter of year $a$.

The estimation of mortality rates, therefore, requires counting the number of people who die as members of a population during a particular period, ${}_s^r D_x^a$, and measuring the total amount of time (in years or quarters) that all the members of the population spend at risk of death (as a member of the population) during that period, ${}_s^r L_x^a$. The `qlifetable` package has been programmed to calculate, for both general and insured populations, the ${}_s^r D_x^a$ and ${}_s^r L_x^a$ statistics, among other figures. By default, these statistics are assumed to be computed by treating individual records that contain dates of births and events; however, when any of these records are not available, some approximations are still possible as long as ${}_s D_x^a$ figures are available. The package includes appropriate functions to facilitate the entire process, ultimately resulting in the construction of four quarterly life tables consistent with a reference annual life table of either a general or insured population.

From now on and without loss of generality, for the rest of the methodological expositions, we assume that we are dealing with data from a general population for which we have detailed information about births, deaths, emigrants, immigrants and population stocks. In the case of a population of insured, policy cancellations and lapses can be interpreted as emigrants, new policy acquisitions as immigrants, population stocks are seen as equivalent to portfolio counts/reviews and there are no births. We rely on the Lexis diagram to explain how the different summary statistics are computed.

The Lexis diagram is a powerful tool for visualizing and analysing demographic data [12]. A Lexis diagram is a type of Cartesian representation used to show the relationship between age and time of demographic events in a population. Sometimes a cohort axis is also made explicit to study how the population evolve over time. The age axis identifies the exact ages of the individuals at the moments of the events. The time (period/calendar) axis locates the exact calendar times in which the events happen. Lifelines are also basic elements of a Lexis diagram. Lifelines are diagonal lines that show how individuals move through different age groups and time periods.

Fig 1 shows a slice of the Lexis diagram that spans two age years (from exact age $x$ to exact age $x+2$) and one calendar year (from 00:00:00 on January 1st of year $a$ to 24:00:00 on December 31st of year $a$, or 00:00:00 on January 1st of year $a+1$). Three panels of the same region are represented in Fig 1 to help graphically stress different features. In each panel, 1×1-year cells have been divided into sixteen 1×1 $(r, s)$-quarter cells to properly visualize the quarterly dimension of the data.

In a Lexis diagram of a general population each personal history begins with birth (lifelines beginning at the base of the diagram, exact age 0; not shown in Fig 1) or with the event of immigration, which we denote by '$\bigcirc$' (example: line $l_3$ in Fig 1-left), and ends with death, symbol '+' (example: line $l_1$ in Fig 1-left), or with emigration, symbol 'x' (example: line $l_5$ in Fig 1-left). Of course, people can also immigrate and emigrate (die) in the same calendar year (example: line $l_4$ in Fig 1-left). People who are still alive as a member of the target population at the end of year $a$ are also represented (examples: lines $l_2$ and $l_3$ in Fig 1-left).

Within each age and calendar year, we identify each 1x1-quarter cell using two coordinates: $r$ and $s$ ($r, s$ = 1, 2, 3, 4) (see the middle panel of Fig 1). The first coordinate, $r$, refers to the age quarter and the second coordinate, $s$, to the calendar (season) quarter. For a person of age $x$ last birthday, an event occurs in the age quarter $r$ and the season quarter $s$ iff (a) her/his exact

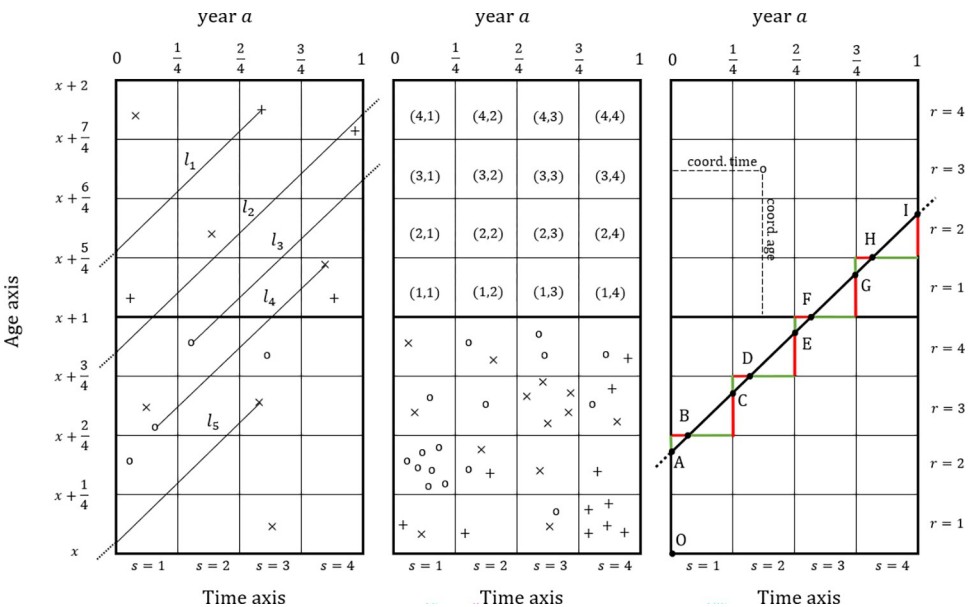

**Fig 1. Excerpt of the Lexis diagram spanning two age years (from exact age *x* to exact age *x*+2) and one calendar year (year *a*).** The left panel displays some lifelines and a schematic representation of some events (deaths +, immigrants ○, emigrants ×). The middle panel labels in the (*r*, *s*) terminology each of the 1x1-quarter cells and presents schematically some more demographic events. The right panel shows a lifeline detailing the time/space the lifeline spans in each (*r*, *s*)-quarter it transits.

age in the moment of the occurrence of the event belongs to $\left[x - \frac{r-1}{4}, x + \frac{r}{4}\right]$ and (b) the time elapsed since the beginning of the calendar year (00:00 AM on 1 January of year *a*) and the moment of the occurrence of the event lies within the interval $\left[\frac{s-1}{4}, \frac{s}{4}\right]$. Generally speaking, calendar quarters can be matched with meteorological seasons [11]. This is relevant because, as a rule, the risk of mortality grows with age and is higher in winter. Indeed, conditional to an integer age, the season has a higher impact on increasing or decreasing the risk of death than the age quarter, with their relationships showing interaction effects [2, 11].

In each (*r*, *s*)-quarter, events of the three types can occur and the number of them occurring in each quarter can be counted. For example, to measure the number of people dying with integer age *x* in the (*r*, *s*)-quarter of year *a*, $_s^r D_x^a$, all that is necessary is to count the number of crosses recorded in the Lexis diagram in that particular surface (example: see quarter (1, 4) in the lower square of Fig 1-middle). In a similar fashion, we can define the number of people immigrating with integer age *x* in the (*r*, *s*)-quarter of year *a*, $_s^r I_x^a$, by counting the number of circles (example: see quarter (2, 1) in the lower square of Fig 1-middle) and the number of people emigrating with integer age *x* in the (*r*, *s*)-quarter of year *a*, $_s^r E_x^a$, by counting the number of blades (example: see quarter (3, 3) in the lower square of Fig 1-middle).

Measuring the contribution of each person to the time exposed to risk in each (*r*, *s*)-quarter of year *a* and age *x*, $_s^r L_x^a$, is more laborious. Indeed, to avoid programming bugs and ensure a systematic computation, we consider thirty-two different scenarios for measuring the time of exposure of deaths and migrants. These scenarios are based on whether the event occurred in the upper or lower triangle of each cell. (Note that each cell can be split into two triangles using a segment of a lifeline). In order to explain how these times are systematically measured, we take as an example the lifeline depicted in the most-right panel of Fig 1, which corresponds to a person who remains alive as a member of the target population during the whole year *a*. The most abstract mathematical explanations can be consulted in [11].

The process starts by computing the exact age of the person at the beginning or the end of the year, which we denote by $y$. Let us assume that it corresponds to the exact age at the beginning of the year: point $A$ in Fig 1-right. From $y$, we calculate two new quantities: the fractional age in years of the individual at the beginning of the year, $coord.age = y - \lfloor y \rfloor$ (i.e., the Euclidean distance between points $O$ and $A$), and the age quarter $r = \lfloor 4\, coord.age \rfloor + 1$ where $A$ is located; $\lfloor \cdot \rfloor$ being the floor function (i.e., the function that for any real number returns the greatest integer number less than or equal to it).

Knowing the age quarter $r$ of the lifeline at the beginning of the year gives us the eight $(r, s)$ quarters that the lifeline transits during year $a$: $(r, 1)$, $(mod_{\bar{4}}(r+1), 1)$, $(mod_{\bar{4}}(r+1), 2)$, $(mod_{\bar{4}}(r+2), 2)$, $(mod_{\bar{4}}(r+2), 3)$, $(mod_{\bar{4}}(r+3), 3)$, $(mod_{\bar{4}}(r+3), 4)$ and $(mod_{\bar{4}}(r+4), 4)$; where $mod_{\bar{4}}(\cdot)$ is the modified modulo 4 operator, which given any integer number returns one plus the remainder of its division by 4 (i.e., an integer number between 1 and 4). In the example in Fig 1-right, the quarters that the depicted lifeline moves through are (2, 1), (3, 1), (3, 2), (4, 2), (4, 3), (1, 3), (1, 4) and (2, 4). Note that the first quarters are located in the 1×1-year cell corresponding to age $x = \lfloor y \rfloor$ and the last quarters in 1×1-year cell of age $\lfloor y \rfloor + 1$, with the transition occurring when $r$ decreases.

Once the ordered $(r, s)$-quarters that the lifeline goes through have been determined, its contribution (in years) to the total time exposed to risk in each quarter it traverses is computed. Each exposure time is equal to the length, divided by $\sqrt{2}/4$, of the segment of the lifeline that is inside the corresponding cell quarter. In the example, these are the distances between points $A$ and $B$, $B$ and $C$, $C$ and $D$, $D$ and $E$, $E$ and $F$, $F$ and $G$, $G$ and $H$ and $H$ and $I$, divided by $\sqrt{2}/4$. Note that after dividing by $\sqrt{2}/4$ these distances (times in years) are equal to the distances of their linked horizontal and vertical coloured segments. For instance, the Euclidean distance between $A$ and $B$ divided by $\sqrt{2}/4$ is equal to both the (minimum Euclidean) distance between $A$ and the line corresponding to age $x + \frac{1}{2}$ and the (minimum Euclidean) distance between $B$ and the line corresponding to time 00:00 AM on 1 January of year $a$. This makes it straightforward to measure all the distances (contributions of the individual to the total time-population at risk in each quarter) as they alternate each of the figures $\frac{r}{4} - coord.age$ and $coord.age - \frac{r-1}{4}$ four times. In the example, the contributions of the lifeline defined by the points $A$ and $I$ to the (2, 1), (3, 1), (3, 2), (4, 2), (4, 3), (1, 3), (1, 4) and (2, 4) quarters are: $0.5 - coord.age$, $coord.age - 0.25$, $0.5 - coord.age$, $coord.age - 0.25$, $0.5 - coord.age$, $coord.age - 0.25$, $0.5 - coord.age$, and $coord.age - 0.25$, respectively.

Finally, in Fig 1-right, an example is also shown of the $coord.age$ and the $coord.time$ corresponding to an immigration event. This allows us to locate the event in the 1x1-year cell. The values of these coordinates are equal to the lengths of the corresponding segments: the times elapsed (in years) since (i) the date of last birthday of the immigrant and (ii) the beginning of the year up to the date of the event, respectively. The first set of functions in the qlifetable package (see subsection 5.1) is designed to calculate summary statistics from detailed individual microdata records, when available, by applying the procedures presented in this section.

## 3. Estimating quarterly life tables

Once ${}_{s}^{r}D_{x}^{a}$ and ${}_{s}^{r}L_{x}^{a}$ statistics have been computed, for $r, s = 1, 2, 3, 4$ and $x = m, m+1, \ldots, M$, in a set of years $a = a_1, \ldots, a_T$ —where $m$ and $M$ denote the minimum and maximum ages in the dataset and $T$ the number of years available—, crude estimates for the (central) annual and quarterly death rates, $m_x^a$ and ${}_{s}^{r}m_x^a$, are straightforwardly attained through Eqs (1) and (2):

$$\hat{m}_x^a = \frac{D_x^a}{L_x^a} \tag{1}$$

$$_s^r\hat{m}_x^a = \frac{_s^rD_x^a}{_s^rL_x^a} \qquad (2)$$

where $D_x^a = \sum_{r,s=1}^4 {}_s^r D_x^a$ and $L_x^a = \sum_{r,s=1}^4 {}_s^r L_x^a$.

At this point, (raw) death probability estimates, $_s^r\hat{q}_x^a$, may be directly derived from the $_s^r\hat{m}_x^a$ estimates and from them (after smoothing the series of probabilities) quarterly life tables constructed. This approach, however, shows a significant drawback [11]: it requires a very large amount of data to reach reliable estimates. As each 1x1-annual cell is divided into sixteen quarters, each $_s^r\hat{m}_x^a$ estimate is based on a sample that has on average the sixteenth part of the size the sample used to estimate $\hat{m}_x^a$. This makes it difficult to use this approach in small and medium-sized life insurance companies, as death rates are not time stationarity [13].

As an alternative, and exploiting the fact that (i) $\hat{m}_x^a$ and $_s^r\hat{m}_x^a$ are in the same scale and (ii) the $\hat{m}_x^a$ estimates are weighted averages of the $_s^r\hat{m}_x^a$ estimates, Pavía and Lledó [11] propose modelling the deviations of the quarterly rates $_s^r\hat{m}_x^a$ around their average values and estimating seasonal-ageing indexes (SAIs) for each age $x$ and pair $(r, s)$, $\gamma_{rs}^{(x)}$, as intermediate instruments. Despite the SAIs not being time stationary either, they evolve over time noticeably slower [14], which makes it safer to combine data from different years.

In particular, Pavía and Lledó [11] suggest a three step approach that consists in (i) attaining initial raw estimates of $\hat{\gamma}_{rs}^{(x)}$ by fitting model (3), (ii) adjusting and smoothing them, $\tilde{\gamma}_{rs}^{(x)}$, and (iii) using them to derive smoothed estimates of quarterly death rates $_s^r\tilde{m}_x = \tilde{m}_x\tilde{\gamma}_{rs}^{(x)}$ from which quarterly life tables are built.

$$log\left(\frac{_s^r\hat{m}_x^a}{\hat{m}_x^a}\right) = log\left(\gamma_{rs}^{(x)}\right) + {}_s^r\varepsilon_x^a \quad for \quad a = a_1, \ldots, a_T \qquad (3)$$

where $_s^r\varepsilon_x^a$ are random disturbances.

Finally, death probabilities for the quarterly tables are approximated using, for instance, the expression $_{\frac{1}{4}}\tilde{q}_{x+\frac{(r-1)}{4}}^{(s)} = \frac{_s^r\tilde{m}_x}{4+\frac{1}{2}_s^r\tilde{m}_x}$, where it is implicitly assumed that individuals who died contribute half a quarter person-year of exposure (an assumption not made when estimating death rates), and the four possible period quarterly life tables, which depend on the season of birth, constructed using the following series of death probabilities:

- $_{\frac{1}{4}}\tilde{q}_0^{(1)}$ , $_{\frac{1}{4}}\tilde{q}_{\frac{1}{4}}^{(2)}$ , $\cdots$ , $_{\frac{1}{4}}\tilde{q}_{x-\frac{1}{4}}^{(4)}$ , $_{\frac{1}{4}}\tilde{q}_x^{(1)}$ , $_{\frac{1}{4}}\tilde{q}_{x-\frac{1}{4}}^{(2)}$ , $_{\frac{1}{4}}\tilde{q}_{x-\frac{2}{4}}^{(3)}$ , $_{\frac{1}{4}}\tilde{q}_{x-\frac{3}{4}}^{(4)}$ , $_{\frac{1}{4}}\tilde{q}_{x-\frac{5}{4}}^{(1)}$ , $\cdots$ .

- $_{\frac{1}{4}}\tilde{q}_0^{(2)}$ , $_{\frac{1}{4}}\tilde{q}_{\frac{1}{4}}^{(3)}$ , $\cdots$ , $_{\frac{1}{4}}\tilde{q}_{x-\frac{1}{4}}^{(1)}$ , $_{\frac{1}{4}}\tilde{q}_x^{(2)}$ , $_{\frac{1}{4}}\tilde{q}_{x-\frac{1}{4}}^{(3)}$ , $_{\frac{1}{4}}\tilde{q}_{x-\frac{2}{4}}^{(4)}$ , $_{\frac{1}{4}}\tilde{q}_{x-\frac{3}{4}}^{(1)}$ , $_{\frac{1}{4}}\tilde{q}_{x-\frac{5}{4}}^{(2)}$ , $\cdots$ .

- $_{\frac{1}{4}}\tilde{q}_0^{(3)}$ , $_{\frac{1}{4}}\tilde{q}_{\frac{1}{4}}^{(4)}$ , $\cdots$ , $_{\frac{1}{4}}\tilde{q}_{x-\frac{1}{4}}^{(2)}$ , $_{\frac{1}{4}}\tilde{q}_x^{(3)}$ , $_{\frac{1}{4}}\tilde{q}_{x-\frac{1}{4}}^{(4)}$ , $_{\frac{1}{4}}\tilde{q}_{x-\frac{2}{4}}^{(1)}$ , $_{\frac{1}{4}}\tilde{q}_{x-\frac{3}{4}}^{(2)}$ , $_{\frac{1}{4}}\tilde{q}_{x-\frac{5}{4}}^{(3)}$ , $\cdots$ .

- $_{\frac{1}{4}}\tilde{q}_0^{(4)}$ , $_{\frac{1}{4}}\tilde{q}_{\frac{1}{4}}^{(1)}$ , $\cdots$ , $_{\frac{1}{4}}\tilde{q}_{x-\frac{1}{4}}^{(3)}$ , $_{\frac{1}{4}}\tilde{q}_x^{(4)}$ , $_{\frac{1}{4}}\tilde{q}_{x-\frac{1}{4}}^{(1)}$ , $_{\frac{1}{4}}\tilde{q}_{x-\frac{2}{4}}^{(2)}$ , $_{\frac{1}{4}}\tilde{q}_{x-\frac{3}{4}}^{(3)}$ , $_{\frac{1}{4}}\tilde{q}_{x-\frac{5}{4}}^{(4)}$ , $\cdots$ .

The second set of functions in the `qlifetable` package (see subsection 5.2) is designed to estimate crude death rates, calculate SAIs, and build quarterly life tables, either by using one of the shortcuts suggested in [2] when microdata events for some or all the variables are missing, or by applying the procedures presented in this section.

## 4. Leap years, times exposed to risk and exact ages at events

In the previous sections, we have implicitly assumed that the lengths of age and calendar (time) years are equal and constant over time. Unfortunately, age and calendar years have

different lengths, and are composed of a different number of days. This is due to the non-divisibility between the Earth's translational and rotational cycles; a discrepancy that gives rise to the need for leap years. There are calendar years of 365 days and 366 days. As a rule, common calendar years have 365 days, with approximately one in every four years having 366 days. To be precise, every year that is a multiple of four is a leap year unless the year is perfectly divisible by 100 but not perfectly divisible by 400, in which case it is not a leap year. For example, the years 1600 and 2000 were leap years, but the years 2100 and 2200 will be common years. However, one complete orbit of the Earth around the Sun takes around 365.249 days. In other words, there is not a perfect correspondence between the length of calendar and orbital years, although a simple pattern is present. Each year in a person´s lifetime will have, on average, approximately 365.25 days. Given this, a possible solution to ensure constant and equal-length years would be to consider both age and calendar years as having 365.25 days. This solution, however, is not operative as the dates of occurrence of events (births, deaths and migrations) are recorded in calendar days.

Fortunately, our computations are performed on a year by year basis, so we only need for both age and calendar years to have the same length in each time year. Hence, as a practical solution, it would be enough to consider the length of both age and calendar years equal to the calendar year in each time year. This is an operative and reasonable solution that satisfactorily solves the problem of how to measure time within each 1x1-year cell once the event is located in the cell. Nevertheless, it does not solve by itself the problem of how to exactly locate events in the Lexis diagram, i.e., how to determine time and age coordinates (see Fig 1-right).

This problem is not present in the case of a time coordinate (*time.coord*), since it is enough to compute the time elapsed from the beginning of the year to the instant of the event. The difficulty comes when measuring the age coordinate (*age.coord*) or, equivalently, the exact age of the individual at the moment of the event, i.e., the time elapsed (in years) between the birth and the moment of the (death/migrant) event. Given that the person usually transits many years between both events, a reasonable solution would be to consider the length of the average year and to compute the exact age as the quotient between (i) the differences in days between the instant of the event and the instant of birth and (ii) 365.25. This option is available in the functions of qlifetable, but it is not the one stated as default; the reason being that, as we can see in Tables 1 and 2, this leads to unexpected and counterintuitive results. For instance, when birth and events happen in the same calendar year, the time elapsed within the 1x1-annual cell will usually be different in both axes and among years. Similarly, the amount of time elapsed between two events that occur at exactly the same point in different years will only be a whole number if the difference in years is divisible by four. The first effect, taking as reference the origin (the start of the age or the year; point *O* in Fig 1-right), is shown in Table 1 and the second one is illustrated employing several examples in Table 2.

**Table 1. Examples of time and age coordinates of the event as a function of the length of the year utilised to calculate the exact age at event.**

| Birth | Event | Time | Age coordinate | |
|---|---|---|---|---|
| date | date | coordinate | Length year: subjective | Length year: constant |
| 2019-01-01 | 2019-06-01 | 0.4150685 | 0.4150685 | 0.4147844 |
| 2020-01-01 | 2020-06-01 | 0.4166667 | 0.4166667 | 0.4175222 |
| 2019-01-01 | 2019-02-01 | 0.0863014 | 0.0863014 | 0.0862423 |
| 2020-01-01 | 2020-02-01 | 0.0860656 | 0.0860656 | 0.0862423 |

Note: The figures are computed considering a baby who is born at the beginning of the year (at time 00:00:00 of January, 1) and migrates/dies at noon of the day of the event. The format of the dates is yyyy-mm-dd.

**Table 2. Examples of exact ages at events as a function of the length of the year utilised to calculate them when births and events happen at exactly the same moment in two different time years.**

| Birth | Event | Exact age at event | |
|---|---|---|---|
| date | date | Length year: constant | Length year: subjective |
| 2019-01-01 00:00:00 | 2023-01-01 00:00:00 | 4.000000 | 4.000000 |
| 2020-01-01 00:00:00 | 2023-01-01 00:00:00 | 3.000685 | 3.000000 |
| 2021-01-01 00:00:00 | 2023-01-01 00:00:00 | 1.998631 | 2.000000 |
| 2022-01-01 00:00:00 | 2023-01-01 00:00:00 | 0.999316 | 1.000000 |
| 2019-02-01 00:00:00 | 2023-02-01 00:00:00 | 4.000000 | 4.000000 |
| 2020-02-01 00:00:00 | 2023-02-01 00:00:00 | 3.000685 | 3.000232 |
| 2021-02-01 00:00:00 | 2023-02-01 00:00:00 | 1.998631 | 2.000000 |
| 2022-02-01 00:00:00 | 2023-02-01 00:00:00 | 0.999316 | 1.000000 |
| 2019-04-01 00:00:00 | 2023-04-01 00:00:00 | 4.000000 | 4.000000 |
| 2020-04-01 00:00:00 | 2023-04-01 00:00:00 | 2.997947 | 2.997941 |
| 2021-04-01 00:00:00 | 2023-04-01 00:00:00 | 1.998631 | 2.000000 |
| 2022-04-01 00:00:00 | 2023-04-01 00:00:00 | 0.999316 | 1.000000 |
| 1949-07-01 00:00:00 | 2023-07-01 00:00:00 | 73.99863 | 74.00000 |
| 1950-07-01 00:00:00 | 2023-07-01 00:00:00 | 72.99932 | 73.00000 |
| 1951-07-01 00:00:00 | 2023-07-01 00:00:00 | 72.00000 | 72.00000 |
| 1952-07-01 00:00:00 | 2023-07-01 00:00:00 | 70.99795 | 70.99862 |

Note: The format of the dates is yyyy-mm-dd hh:mm:ss.

These anomalies can be almost completely removed (i.e., solved for the majority of the computations/persons) by adopting a quite counterintuitive assumption: the hypothesis that the lengths of the age years are relative, subjective. In particular, the assumption that they depend on the moment of birth of the person and on the moment of occurrence of the event (in general, the moment in which we are interested in for measuring the exact age). Specifically, we assume that the length of the age year of a moment-of-birth and moment-of-event pair is equal to the weighted average of the length of the years the person lives between both moments with each year weight equal to the proportion of that year the person has transited. Mathematically, let $l_a$ be the length of the calendar year $a$ and $f_a$ the fraction of year $a$ enclosed between the moments of birth, $t_b$, and event, $t_e$. We define the length of the year associated with this birth-event pair through Eq (4) and the exact age at event through Eq (5):

$$l_{(b,e)} = \frac{\sum_{a=a_b}^{a_e} l_a f_a}{\sum_{a=a_b}^{a_e} f_a} \tag{4}$$

$$exact.age.at.event = \frac{d(t_b, t_e)}{l_{(b,e)}} \tag{5}$$

where $a_b$ and $a_e$ represent the calendar years where birth and event happen, respectively, and $d(t_b, t_e)$ denotes the distances in days between $t_b$ and $t_e$.

The great advantage of using a non-constant (person-dependent) length of year to compute exact ages at events is congruence of distances between axes when estimating time exposed (non-exposed) to risk in each year. This is clearly observed if we measure the age and time coordinates corresponding to the migration/death event of a baby who is born at the beginning of the year and also migrates/dies in the same year. As can be seen in Table 1, when the length of year employed to calculate the exact age at event is calculated as in Eq (4), time and

age coordinates coincide, something that does not happen when the length of the year used to calculate the exact age at event is taken as constant.

This procedure to compute the exact age of a person at the moment of the event also has another desirable result. As a rule, the time elapsed between two events that occur at exactly the same calendar instant in two different time years is a whole number. The only deviations to this rule happen when the birth event occurs in a leap year. This matches with how we measure ages in common life. However, when a constant length of years is employed to calculate exact ages at events, this only happens if the distance between both moments is exactly four years. These features are illustrated with several examples in Table 2.

## 5. The qlifetable functions

The qlifetable package has been developed to implement the methodological solutions described in sections 2 to 4 and is available, under the Eclipse Public License (EPL), on the Comprehensive R Archive Network (CRAN) at <https://CRAN.R-project.org/package= qlifetable>. Thus, it can be installed and loaded into a current R session using the code:

R> install.packages("qlifetable")

R> library("qlifetable")

The package incorporates two sets of integrated functions for constructing quarterly life tables. The first set of functions (subsection 5.1) is tailored to compute necessary intermediate datasets of summary statistics from microdata records. The second set of functions (subsection 5.2) completes the process by offering solutions for estimating crude death rates, computing SAIs, and building quarterly life tables. These functions can be used to implement both the full-data approach outlined in [11] as well as the less data-demanding shortcuts proposed in [2].

The inputs for the second set of functions are datasets of summary aggregate statistics, which can either be provided directly by the user or derived through the first set of functions from detailed microdata event datasets. The complete process is graphically summarised in Fig 2. Fig 2 schematically illustrates the pipeline flow to derive final quarterly life tables from either

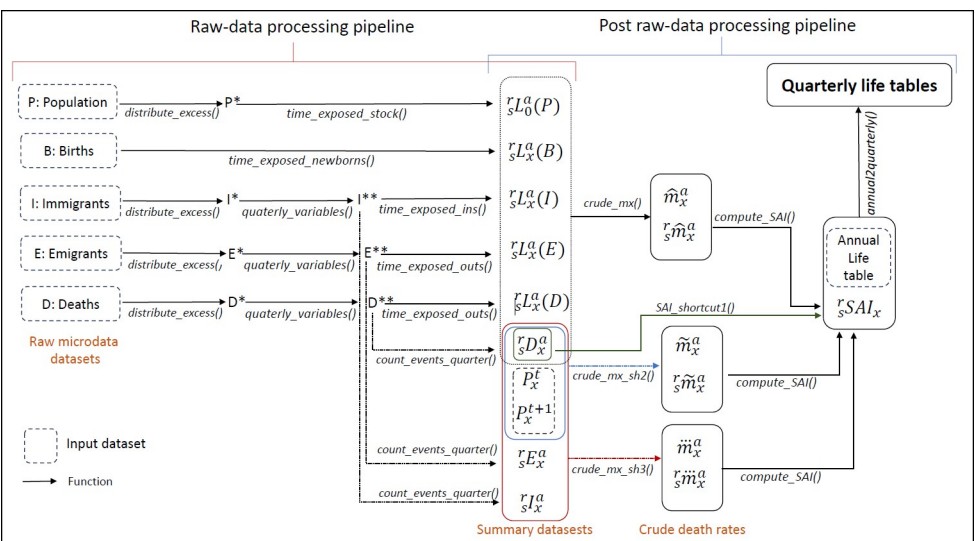

**Fig 2. Schematic representation of the pipeline flow for deriving quarterly life tables from detailed microdata or summary statistics.**

detailed microdata or summary statistics, with all intermediate outputs consisting of quantities of interest that can be analysed in depth or graphically plotted.

## 5.1 Functions to deal with microdata

Table 3 offers a summary and concise description of the functions exported in the package for dealing with microdata. These functions facilitate the generation of summary datasets from extensive microdata records containing dates of births and events of a (general or insured) population. The synthetic statistics generated by these functions at the end of the raw-data processing pipeline (see the left-side of Fig 2) include the number of deaths and total times exposed to risk for each combination of integer age, age-quarter, and season-quarter.

A list with the main arguments of the functions described in Table 3 is presented in Table 4. The list embraces arguments for dates of birth and events as well as indicators about how these must be considered within the day. The main inputs for these functions are vectors containing dates of births and dates of events in either the "yyyy-mm-dd" or "yyyy-mm-dd hour:min:secs" format (e.g., "2016-01-20"). Usually the exact moment when births or events happen within the day is unknown and it is almost never recorded even if known. So, to avoid systematic biases qlifetable functions allow instants within days to be randomly selected. This is done by leaving the arguments random.b and random.e (see Table 4) as default. Table 4 also contains a brief description of the arguments and lists the functions where these arguments are employed.

In addition to the arguments listed in Table 4, some of the functions in Table 3 have other relevant arguments. The most important arguments not included in Table 4 are the year and

**Table 3. Description of the functions in qlifetable for dealing with microdata.**

| Function | Description |
|---|---|
| coord_age | Computes the coordinates corresponding to the age axis of a population at the dates of events. |
| coord_time | Computes the coordinates corresponding to the calendar axis of a vector of events. |
| count_events_quarter | Computes for each integer age $x$ the number of events (of a given type) that happen in a population in each $(r, s)$-quarter. |
| distribute_excess | Randomly distributes the excess of births artificially recorded in a given day, often the first of January, due to usual immigration policies. |
| exact_age | Computes the exact ages in years that the members of a population have when events happen to them, or in a particular moment in time. |
| plot.qlifetable | A graphical method to plot, in a 4x4 raster, the sum across (a subset of) ages by $(r, s)$-quarter of the number of events or the total time exposed to risk in the population. |
| quarterly_variables | Generates a dataset containing age and time coordinates as well as other related statistics (such as the $r$ and $s$ quarter or the age $x$) of a vector events of a population. This works as input of the next two functions. |
| time_exposed_ins | Computes for each integer age $x$ and each combination of an age- and season-quarter, $(r, s)$, the total person-years that a population of migrants or deaths would have been (was) at risk of dying from the occurrence of the events up to the end of the calendar year. |
| time_exposed_outs | Computes for each integer age $x$ and each combination of an age- and season-quarter, $(r, s)$, the total person-years that a population of deaths or migrants was (would have been) at risk of dying from the beginning of the year up to the occurrence of the events. |
| time_exposed_newborns | Computes for the integer age 0 and each season-quarter, $(r, s)$, the total person-years that a population of new-borns was (would have been) at risk of dying from the instant of their birth up to the end of the calendar year in which they are born. |
| time_exposed_stock | Computes for each integer age $x$ and each combination of an age- and season-quarter, $(r, s)$, the total person-years that a stock of population counted at the beginning (at the end) of the calendar year would be (has been) at risk of dying during the year. |

**Table 4. Description of the main arguments of the functions in qlifetable.**

| Argument | Description | Functions involved |
|---|---|---|
| date.birth | A character vector with the dates of birth in format either "yyyy-mm-dd" or "yyyy-mm-dd hour:min:secs" (for instance, "2016-01-20 12:00:00") of a population. | coord_age, distribute_excess, exact_age, quarterly_variables, time_exposed_newborns, time_exposed_stock |
| date.event | A character vector with the dates of events in format either "yyyy-mm-dd" or "yyyy-mm-dd hour:min:secs" (for instance, "2016-01-20 12:00:00") of a population. | coord_age, coord_time, distribute_excess, exact_age, quarterly_variables, time_exposed_stock |
| random.b | A 'TRUE/FALSE' argument indicating whether the exact moment ("hour:min:secs") when the birth occurs within the day is randomly selected. Default TRUE. | coord_age, exact_age, quarterly_variables, time_exposed_newborns, time_exposed_stock |
| random.e | A 'TRUE/FALSE' argument indicating whether the exact moment ("hour:min:secs") when the event occurs within the day is randomly selected. Default, TRUE. | coord_age, coord_time, exact_age, quarterly_variables |
| constant.age. year | A 'TRUE/FALSE' argument indicating whether the length of the year should be constant, 365.25 days, or variable, depending on the time lived for the person in each year since her/his dates of birth and event. Default, FALSE. | coord_age, exact_age, quarterly_variables, time_exposed_stock |
| x | A data frame output of the quarterly_variables function. | count_events_quarter, time_exposed_ins, time_exposed_outs |

type arguments of the function time_exposed_stock. Given that this function computes for each integer age $x$ and each combination of an age- and season-quarter, $(r, s)$, the total time exposed at risk (in years) of a (stock) population of survivors (expected survivors) during a given year (denoted as $_s^r L_x^a$ in Fig 2), the function needs to be informed about the year the stock of population was recorded (which is assumed to be at 00:00:00 of January, 1, or 24:00:00 of December, 31, of that year) and about whether the time exposed must be computed from the beginning of the year (type = "forward") or from the end of the year (type = "backward"). If type = "forward", it is assumed that the population counting has been performed at the beginning of the year of interest while, if type = "backward", it is assumed that the population counting has been performed at the end of the year of interest. At the end of the raw-data processing pipeline, among other summary statistics, the package generates datasets of the exposure times corresponding to the stock of population, immigrants, emigrants, deaths and births. These are represented in the notation of Fig 2 as $_s^r L_x^a(P)$, $_s^r L_x^a(I)$, $_s^r L_x^a(E)$, $_s^r L_x^a(D)$ and $_s^r L_0^a(B)$, respectively. These datasets are then combined in different ways, depending on the value of type argument, to obtain $_s^r L_x^a$, as detailed in the next subsection.

Another function that deserves more details is the function distribute_excess. Demographic data available in official statistical agencies are not perfect, even in well-developed countries [15, 16]. These imperfections affect the distributions of dates of births due mainly to the policies implemented by immigration/census agents. In many countries, when an immigrant is enrolled in the population register and s/he is unsure of their exact date of birth, it is administratively recorded as 1 January. This results in an artificial surge in the number of births recorded on that day and the impact of this is a bias in the distributions of births on both stocks of population and migration statistics. The distribute_excess function has been created to randomly assign dates of birth to a random subsample of size equal to the excess of people recorded in any particular day; by default, the first of January. The argument maximum.excess, set by default at 50, is used to inform the function about the percentage of births registered above the average in a given year that must be exceeded on the target day to identify an artificially recorded excess of births.

The function with more arguments in this set is the function plot.qlifetable. This function is a convenient method for plotting a quarterly summary of events or times of exposition. The method uses the ggplot2 package [17] and offers users numerous customization options through its various arguments. (Please see the documentation of the package for details.) The resulting plot can be saved as a ggplot2 object and manipulated further. Note that for this function to run, the ggplot2 package needs to be installed.

## 5.2. Functions to estimate quarterly death rates and SAIs and build quarterly life tables

Having described the functions in Table 3 and their arguments, we now briefly detail how to use the functions presented in Table 5 to eventually build quarterly life tables, as proposed in [2, 11]. This involves properly combining datasets of summary statistics, which may or not have been derived from microdata records by using the set of functions detailed in subsection 5.1.

The complete post raw-data processing pipeline (see the right-side of Fig 2) encompasses computing tables of estimates of crude (central) quarterly death rates, ${}_s^r\hat{m}_x^a$ (except when the shortcut-1 by [2] is employed), for several years; estimating seasonal-ageing indexes (SAIs); and, applying the estimated SAIs to the annual life table of the target population. Tables of ${}_s^r\hat{m}_x^a$ estimates can be generated using the functions crude_mx, crude_mx_sh2 and crude_mx_sh3, with the particular summary datasets and function to be employed depending on the approach followed: either the full-data method described in [11] or shortcut-2 or shortcut-3 from [2]. SAIs are obtained employing the function compute_SAI using outputs of the crude_mx-family functions as inputs, except when shortcut-1 in [2] is employed, as they are computed directly from datasets of death statistics, ${}_s^r D_x^a$, with the aid of the SAI_shortcut_1 function. Finally, quarterly life tables are attained through the function annual2quaterly by applying the estimated SAIs to the annual table. In Table 5, one also can find the plot.SAI function, which is a convenient method for plotting an output of either compute_SAI or SAI_shortcut_1. This method shares with plot.qlifetable the requirement of having the ggplot2 package installed, along with all its properties.

We begin by detailing how estimates of crude (central) quarterly death rates are obtained when the full-data approach (the crude_mx function) is employed. Subsequently, we elaborate on how these estimates are calculated in simpler scenarios, where either crude_mx_sh2 or crude_mx_sh3 is utilised.

The estimation of ${}_s^r\hat{m}_x^a$ requires the ${}_s^r D_x^a$ and ${}_s^r L_x^a$ statistics, which, in the full-data approach, are attained by properly combining the outputs (or similar objects) of the functions described in Table 3. Specifically, assuming we are dealing with a general population and are in a full-data scenario, denoting by $P$ the micro-records corresponding to a stock of the population counted either at the beginning or at the end of the year, and by $E$, $I$, $D$ and $B$ the micro-records corresponding to emigrants, immigrants, deaths and births recorded during the year, the number of deaths and person-years lived in each $(r, s)$-quarter by age $x$ are attained as follows. On the one hand, ${}_s^r D_x^a$ statistics are obtained by sequentially applying the quarterly_variables

**Table 5. Description of the functions in qlifetable for building quarterly life tables.**

| Function | Description |
| --- | --- |
| annual2quaterly | Estimates the four quarterly life tables associated to an annual life table given the annual life table and a set of SAIs estimates. |
| compute_SAI | Estimates SAIs from a set of quarterly tables of crude rates estimates corresponding to several years. |
| crude_mx | Computes the estimates of quarterly crude death rates from datasets of quarterly summary statistics. |
| crude_mx_sh2 | Computes from datasets of quarterly summary statistics the estimates of quarterly crude death rates applying the shortcut-2 in [2]. |
| crude_mx_sh3 | Computes from datasets of quarterly summary statistics the estimates of quarterly crude death rates applying the shortcut-3 in [2]. |
| plot.SAI | A method for plotting a graphical representation of the dataset of seasonal-ageing indexes (SAIs) obtained using the functions compute_SAI or SAI_shortcut_1. |
| SAI_shortcut_1 | Estimates SAIs from a set of quarterly datasets of deaths by applying shortcut-1 in [2]. |

and count_events_quarter functions to the *D* dataset (see Fig 2). On the other hand, the calculation of the total time exposed to risk depends on whether the stock of population is referenced at the beginning or at the end of the year of interest.

If the date of reference of the data in *P* is the beginning of the year (case type = "forward"), the formula is: ${}_{s}^{r}L_{x}^{a}$ = time_exposed_stock(*P*) + time_exposed_ins(*I*)–time_exposed_ins(*E*)– time_exposed_ins(*D*) + time_exposed_newborns(*B*). Equivalently, using the notation in Fig 1, ${}_{s}^{r}L_{x}^{a} = {}_{s}^{r}L_{x}^{a}(\text{P}) + {}_{s}^{r}L_{x}^{a}(\text{I}) - {}_{s}^{r}L_{x}^{a}(\text{E}) - {}_{s}^{r}L_{x}^{a}(\text{D}) + {}_{s}^{r}L_{0}^{a}(\text{B})$. Note that the pipeline description in Fig 2 assumes that the stock of population is referenced at the beginning of the year of interest. If the stock of population is referenced at the end of the year of interest (case type = "backward"), the method for computing the total time exposed varies slightly. Specifically, ${}_{s}^{r}L_{x}^{a}$ = time_exposed_- stock(*P*)–time_exposed_outs(*I*) + time_exposed_outs(*E*) + time_exposed_outs(*D*). In the above expressions, before applying the functions of time exposure, the *I*, *E* and *D* datasets need to be previously *transformed* using the quarterly_variables function.

The above equations to compute ${}_{s}^{r}L_{x}^{a}$ are internally integrated into the crude_mx function. Therefore, to estimate quarterly crude death rates, the user only needs to input summary datasets, likely generated from the microdata using the quarterly_variables, count_events_quarter and time_exposed-family functions, and to declare the type argument (either "forward" or "backward"). The main inputs of the crude_mx function are datasets containing ${}_{s}^{r}D_{x}^{a}$ and ${}_{s}^{r}L_{x}^{a}(\cdot)$ statistics. These datasets consist of four columns. The first three columns indicate the quarter corresponding to the statistic displayed in the fourth column, which represents either the number of deaths or the time of exposure. Specifically, the first column refers to the age (*x*), the second to the age-quarter (*r*) and the third to the calendar-quarter (*s*).

The estimation of ${}_{s}^{r}\hat{m}_{x}^{a}$ tables using crude_mx_sh2 and crude_mx_sh3 is simpler. In the first case, only integer-age stocks of populations corresponding to the beginning and the end of the target year and a dataset as generated by the count_events_quarter function when applied to the D dataset are needed. In the second case, in addition to the inputs demanded by crude_mx_sh2, the function crude_mx_sh3 also requires datasets as generated by the count_events_quarter function when applied to the I and E datasets. The structure of the datasets generated by the count_events_quarter function is as described for the inputs of the crude_mx function. In contrast, the integer-age stocks of population dataset is a data frame with two columns: the first column refers to age, and the second indicates the number of people in the population corresponding to that age.

The outputs produced by the functions of the crude_mx-family serve as inputs for the compute_SAI function. This function requires at least two sets or tables of (crude) death rate estimates to calculate SAIs. In the case of applying shortcut-1 of [2], SAIs are directly estimated from death statistics, ${}_{s}^{r}D_{x}^{a}$, without the need for estimates of crude death rates, ${}_{s}^{r}\hat{m}_{x}^{a}$. Hence, the SAI_shortcut_1 function basically only requires at least two datasets of summary statistics of deaths.

The process concludes with the utilisation of the function annual2quaterly, which takes as inputs the reference annual life table and a dataset of SAIs estimated by either the compute_- SAI or SAI_shortcut_1 functions. The reference annual life table consist of a two- or three-column dataset, where the first column represents age and the second column contains either $m_x$ rates or $q_x$ probabilities. If death probabilities ($q_x$) are used, the reference annual life table may include an optional third column representing the average number of years lived for those dying with age *x*, $a_x$. If this last column is missing, $a_x$ is assumed to be constant and equal to 0.5.

Obviously, the complete process could have been condensed into one or two steps by creating a more general function that performs all (or many of) the steps internally and only

required as inputs the raw microdata or the summary datasets. However, based on our research experience, we have found many of the intermediate results to be useful. Therefore, in our view, having them available separately and the ability to compute them individually benefits the analyst.

## 6. qlifetable in action

In this section, we illustrate how qlifetable works. In subsection 6.1, we demonstrate the creation of summary datasets of microdata events by applying some of its functions to the samples of demographic microdata included in the package. In subsection 6.2, we showcase how quarterly datasets of crude death rates, SAIs, and quarterly life tables can be easily obtained from summary datasets using some of the functions described in Table 5. To do so, we utilise some of the datasets available in the DEMOSPA0521 database created by [18] after summarising more than 850 million microdata demographic records from Spain.

### 6.1. Summarising microdata records

The objects pop_2006, birth_2006, emi_2006, immi_2006, and death_2006, which are available in qlifetable, contain a sample composed of half of the demographic events recorded for the male population of Comunitat Valenciana (one of the seventeen Spanish regions) in 2006. Since the dates in these datasets do not provide information on the exact moments within the days when events occurred, to prevent potential systematic biases, we have set the default values for the arguments random.b and random.e, and have randomly selected a moment within the day for each event. The statistics of population are referenced as January, 1 2006.

First, we apply the distribute_excess function, with default options, in order to randomly distribute the potential excesses of men registered in each calendar year as born on the first of January. Then, we apply the quarterly_variables function to summarise in a more useful way the death, emigration and immigration events. This chunk of code ends showing an extract of a typical dataset produced by the quarterly_variables function in Fig 3. To make computations reproducible we start by fixing the random seed.

```
R> set.seed(2311)
R> data(pop_2006)
R> pop_2006$date.birth <- distribute_excess(pop_2006$date.birth)
R> data(immi_2006)
R> immi_2006$date.birth <- distribute_excess(immi_2006$date.birth,
R> + date.event = immi_2006$date.immi)
R> data(emi_2006)
R> emi_2006$date.birth <- distribute_excess(emi_2006$date.birth,
R> + date.event = emi_2006$date.immi)
R> data(death_2006)
R> deaths <- quarterly_variables(date.birth = death_2006$date.birth,
R> + date.event = death_2006$date.death)
R> immi <- quarterly_variables(date.birth = immi_2006$date.birth,
```

```
  coord.age coord.time age.last.birthday exact.age.at.event quarter.age quarter.calendar year
1 0.2931507  0.5767123                89           89.29315           2                3 2006
2 0.2493151  0.5630137                79           79.24932           1                3 2006
3 0.3616438  0.8342466                71           71.36164           2                4 2006
4 0.3232877  0.9904110                60           60.32329           2                4 2006
5 0.2545213  0.1547945                97           97.25452           2                1 2006
6 0.6181413  0.2397260                73           73.61814           3                1 2006
```

**Fig 3. Excerpt of a typical dataset produced by the quarterly_variables function.**

```
R> + date.event = immi_2006$date.immi)
R> emi <- quarterly_variables(date.birth = emi_2006$date.birth,
R> + date.event = emi_2006$date.emi)
R> head(deaths)
```

As can be observed, the quarterly_variables function generates a data frame of seven columns that summarises when the events occur, perfectly placing them on the Lexis diagram.

Secondly, we apply the function count_events_quarter to the newly created object deaths in order to compute how many deaths are registered in each $(r, s)$-quarter by age, ${}^r_s D^a_x$, to then show an excerpt of its output (see Fig 4): a data frame with four columns informing on the number of events that occur in each 1x1-quarter cell.

```
R> deaths.rc <- count_events_quarter(deaths)
R> deaths.rc[1000:1005,]
```

Thirdly, we utilise the different time_exposed_ functions to calculate the contribution to the times at risk of all individuals in each dataset, stratified by $(r, s)$-quarter and age, and aggregate all these contributions, taking into account that we are in a type = "forward" case for population. Note that not all the data frames created by these functions have the same number of rows, as each of them spans from age 0 to the maximum age presented in the corresponding microdata records. This must be taken into account when aggregating all the times of exposition. This precaution is unnecessary when using the crude_mx_ functions, as they have been internally programmed to handle this issue.

```
R> t.immi <- time_exposed_ins(immi)
R> t.death <- time_exposed_ins(deaths)
R> t.emi <- time_exposed_ins(emi)
R> t.pop <- time_exposed_stock(date.birth = pop_2006$date.birth,
R> + year = 2006, type = "forward")
R> data(birth_2006)
R> t.birth <- time_exposed_newborns(date.birth = birth_2006$date.birth)
R> t.total <- t_pop
R> t.total$time.exposed[1:1728] <- t.total$time.exposed[1:1728] +
R> + t.immi$time.exposed -
R> + t.emi$time.exposed -
R> + t.death$time.exposed
R> t.total$time.exposed[1:16] <- t.total$time.exposed[1:16] +
R> + t.birth$time.exposed
```

Finally, we present a graphical representation of deaths and times exposed to risk aggregated across ages (see Fig 5).

```
R> plot.deaths <- plot.qlifetable(deaths.rc, decimal.digits = 0, show.plot = FALSE)
R> plot.exposed <- plot.qlifetable(t.total, show.plot = FALSE)
```

| | age | quarter.age | quarter.calendar | number.events |
|---|---|---|---|---|
| 990 | 61 | 4 | 2 | 8 |
| 991 | 61 | 4 | 3 | 8 |
| 992 | 61 | 4 | 4 | 9 |
| 993 | 62 | 1 | 1 | 9 |
| 994 | 62 | 1 | 2 | 11 |
| 995 | 62 | 1 | 3 | 10 |

**Fig 4. Excerpt of a typical dataset produced by the count_events_quarter function.**

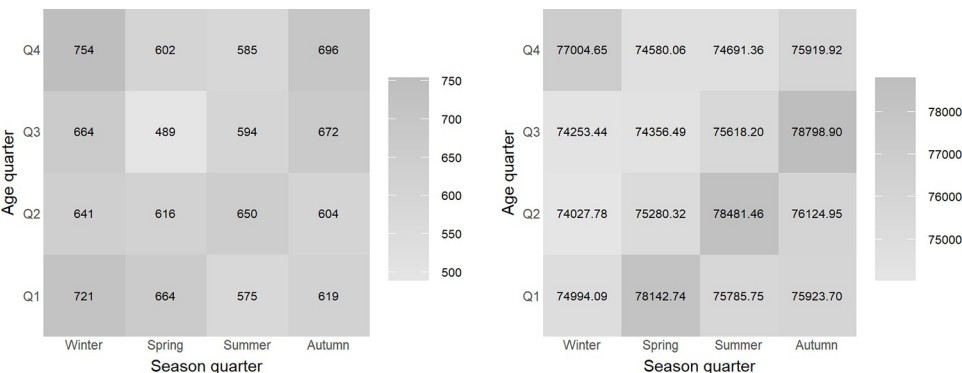

**Fig 5. Graphical representation of deaths and times exposed to risk for each ($r$, $s$)-quarter aggregated across ages corresponding to the datasets available in the qlifetable package.**

## 6.2. Estimating crude death rates, SAIs and quarterly life tables

DEMOSPA0521 [7] is a database composed of fifteen files that summarises approximately 868 million demographic records from Spain, covering the period from 2005 to 2021, in terms of intra-annual distributions. Among other datasets, this database includes quarterly summary statistics of the total time exposed to risk for residents and newborns, as well as both exposure and non-exposure times for deceased individuals and migrants, calculated using the appropriate time_exposed_ functions. These statistics are broken down by integer-age, sex, and year and are calculated for residents assuming a type = "forward" scenario. The database also contains a dataset detailing the number of deaths recorded in Spain by sex and age for each combination of age- and season-quarter in each year. We leverage these datasets to demonstrate how some of the functions in Table 5 can be used to construct a set of quarterly life tables.

As a running example, we focus on the female population for the years 2017 and 2018, for which we estimate quarterly crude death rates. Subsequently, we calculate their associated seasonal-ageing indexes (SAIs) and apply them to the *PASEM2019_second_order* life table (BOE, 2020). This life table is officially recommended and widely utilised by the Spanish insurance industry for reserve calculations, covering both life insurance risk and savings products.

Unfortunately, we cannot directly use the datasets in DEMOSPA0521 as they are; we need first to construct objects similar to those generated by the functions described in Table 3. Although the relevant data for our analysis were produced using the qlifetable functions, the consolidation processes involved in building DEMOSPA0521 required merging datasets and making modifications. On one hand, for logistical purposes, datasets were organised into long table formats where data from different years and both sexes were collected in the same files. On the other hand, for presentation purposes, the labels used to denote age- and season-quarters were changed to Q1 to Q4 and Winter to Autumn, respectively. Since these data wrangling tasks are auxiliary and not part of the package's usage, we have moved them to an S1 Annex. The code presented below continues from where the one in the S1 Annex ends.

First, the function crude_mx is applied to estimate quarterly datasets of crude death rates using type = "forward", as we are considering a full-data scenario where stocks of population are available at the beginning of the year. The code snippet concludes by displaying an extract of a typical dataset produced by a function of the crude_mx-family (see Fig 6).

```
R> mx17 <- crude_mx(time.stock = t.p17, events.death = d17, time.death = t.d17,
R> + time.outs = t.e17, time.ins = t.i17, time.birth = t.b17,
R> + type = "forward")
```

```
       age quarter.age quarter.calendar  exposed deaths         mx
1000    62           2                4 17365.75     78 0.004491600
1001    62           3                1 16984.47     83 0.004886817
1002    62           3                2 16704.03     70 0.004190606
1003    62           3                3 17778.94     63 0.003543519
1004    62           3                4 18273.34     74 0.004049615
1005    62           4                1 18000.51     88 0.004888750
```

**Fig 6. Excerpt of a typical dataset produced by the crude_mx function.**

R> mx18 <- crude_mx(time.stock = t.p18, events.death = d18, time.death = t.d18,

R> + time.outs = t.e18, time.ins = t.i18, time.birth = t.b18,

R> + type = "forward")

R> mx18[1000:1005,]

As can be observed in Fig 6, the crude_mx function generates a data frame of six columns that summarises total times of exposure, number of deaths and crude estimate of death rates, perfectly placing them on the Lexis diagram. The crude_mx_sh2 and crude_mx_sh3 functions yield data frames with identical structure.

After obtaining crude estimates of central death rates for at least two years, SAIs can be estimated using the function compute_SAI. Note that when calling this function, we have included the argument margins = TRUE to compute not only the age-season interaction SAIs but also the marginal ones. The code chunk concludes by plotting the estimated SAIs for the age range from 50 to 100 (see Fig 7).

R> SAIs <- compute_SAI(mx17, mx18, margins = TRUE)

R> plot(SAIs, min.age = 50, max.age = 100)

Finally, the process of building a set of quarterly life tables linked to an annual (general or insured) population life table is completed by applying the estimated SAIs to it. This is done with the help of the function annual2quaterly, as the following code snippet shows. The code chunk ends by displaying an exceprt of the quarterly tables generated using the PASEM2019_-second_order table [19] for women, which is recommended by the Spanish insurance regulator for reserving risk-life insurance products for women.

R> annual.lt <- read.csv("https://links.uv.es/kMB5JRt", sep = ";")

R> annual.lt <- annual.lt[annual.lt$sex = = "Women", c("age", "qx")]

R> quarterly.lt <- annual2quarterly(annual.lt, SAIs)

R> quarterly.lt[250:255,]

As can be observed in Fig 8, the annual2quarterly function generates a data frame of ten columns containing figures for $m_x$ and $q_x$ as a function of the quarter of birth (by columns) for each quarter of age (by rows).

## 7. Concluding remarks

The advent of big data and the revolution of computer power has sparked the development of a novel methodology for estimating sub-annual death probabilities, creating new opportunities for the insurance industry and potentially impacting on the management of pension funds and social security systems. This innovative approach leverages the vast amount of detailed information contained in millions of microdata records to create seasonal-ageing indexes (SAIs), which can be used to derive, considering simultaneously both time dimensions (age and calendar), sub-annual (quarterly) life tables from annual ones. It is important to note, however, that reasonable approximations can also be obtained by processing just a few thousand summary aggregated statistics.

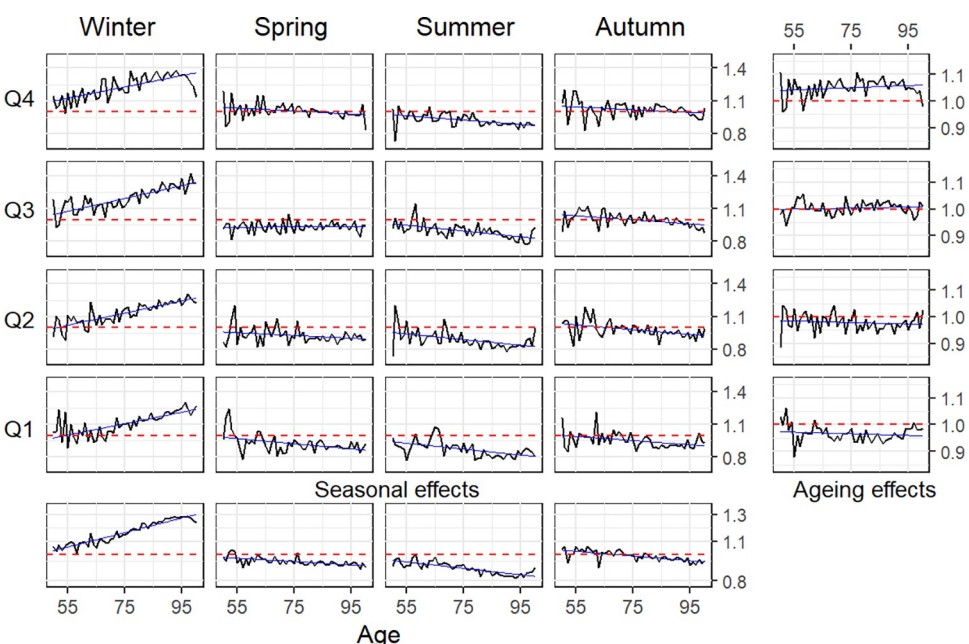

**Fig 7. Graphical representation of estimated Seasonal-Ageing Indexes (SAIs) obtained with the qlifetable package.**

This paper presents the qlifetable package. This instrumental package aids in the tasks of estimating annual and quarterly death rates and building life tables, including deriving new sub-annual tables from referenced annual tables using SAIs. It achieves this by summarising general or insured population microdata of dates of births and events [11] or by directly utilising available summary aggregated datasets [2]. The process of calculating all the required statistics to estimate crude quarterly death rates and SAIs and deriving new quarterly tables is illustrated using some real datasets. The paper also presents the mathematical formulas programmed in the several functions included in qlifetable, as described in the different methodological sections of the paper.

The main limitation in applying the methodology and using the package lies in the minimum data requirements imposed on users. Users need at least age-calendar quarterly death statistics, $_s^r D_x^a$, for at least two years. This requirement may pose a challenge in low- and some middle-income countries, where statistical systems are not sufficiently developed, and microdata detailed records or sub-annual summary statistics are not available. Nevertheless, we believe that this package will become increasingly useful. On the one hand, we expect that more countries will progressively collect and provide detailed demographic information. On the other hand, insurance companies already possess the necessary information in their portfolios.

As a main methodological contribution, this paper shows the need to work with person-dependant age-year lengths in order to guarantee congruency when measuring times of death

| | age | quarter.age | mx.quarter.birth.1 | mx.quarter.birth.2 | mx.quarter.birth.3 | mx.quarter.birth.4 | qx.quarter.birth.1 | qx.quarter.birth.2 | qx.quarter.birth.3 | qx.quarter.birth.4 |
|---|---|---|---|---|---|---|---|---|---|---|
| 250 | 62 | 2 | 0.002290256 | 0.002262976 | 0.002384487 | 0.002916688 | 0.0005724001 | 0.0005655840 | 0.0005959442 | 0.0007289063 |
| 251 | 62 | 3 | 0.002437615 | 0.002401411 | 0.002897606 | 0.002486377 | 0.0006092182 | 0.0006001725 | 0.0007241393 | 0.0006214011 |
| 252 | 62 | 4 | 0.002511709 | 0.003032169 | 0.002385585 | 0.002397533 | 0.0006277301 | 0.0007577549 | 0.0005962186 | 0.0005992037 |
| 253 | 63 | 1 | 0.002991231 | 0.002424561 | 0.002369408 | 0.002576471 | 0.0007475283 | 0.0006059567 | 0.0005921767 | 0.0006439105 |
| 254 | 63 | 2 | 0.002479774 | 0.002442840 | 0.002581515 | 0.003134573 | 0.0006197514 | 0.0006105236 | 0.0006451707 | 0.0007833363 |
| 255 | 63 | 3 | 0.002614822 | 0.002599956 | 0.003134282 | 0.002677369 | 0.0006534919 | 0.0006497777 | 0.0007832635 | 0.0006691182 |

**Fig 8. Excerpt of a typical dataset produced by the annual2quarterly function.**

risk exposure in both Lexis diagram time dimensions. This is a surprising result that compels the observation of time as a relative construct in the disciplines of demography and actuarial science.

## Supporting information

**S1 Annex. R code for adapting datasets in DEMOSPA0521 for use with qlifetable functions.**
(DOCX)

## Acknowledgments

The authors wish to thank Marie Hodkinson for revising the English of the paper and two anonymous reviewers and an associate editor for valuable comments and suggestions.

## Author Contributions

**Conceptualization:** Jose M. Pavía.

**Data curation:** Josep Lledó.

**Formal analysis:** Jose M. Pavía.

**Funding acquisition:** Jose M. Pavía, Josep Lledó.

**Investigation:** Jose M. Pavía, Josep Lledó.

**Methodology:** Jose M. Pavía.

**Project administration:** Jose M. Pavía.

**Resources:** Jose M. Pavía.

**Software:** Jose M. Pavía, Josep Lledó.

**Supervision:** Jose M. Pavía, Josep Lledó.

**Validation:** Jose M. Pavía, Josep Lledó.

**Visualization:** Jose M. Pavía, Josep Lledó.

**Writing – original draft:** Jose M. Pavía, Josep Lledó.

**Writing – review & editing:** Jose M. Pavía.

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
