## [Decision Letter · Decision Letter 0]

14 Oct 2024

PONE-D-24-36834qlifetable: An R package for constructing quarterly life tablesPLOS ONE

Dear Dr. Pavía,

Thank you for submitting your manuscript to PLOS ONE. After careful consideration, we feel that it has merit but does not fully meet PLOS ONE’s publication criteria as it currently stands. Therefore, we invite you to submit a revised version of the manuscript that addresses the points raised during the review process.

This is a very interesting and relevant paper, however it needs to work on different sections:see reviewers comments organization of the material should be revisedprovide more information on the functionalities of the packageimprove discussion on the limitations and applicabitymore detail on how to organize data and other information

We look forward to receiving your revised manuscript.

Kind regards,

Bernardo Lanza Queiroz, Ph.D

Academic Editor

PLOS ONE

Journal Requirements:

 This work was supported by Generalitat Valenciana, Conselleria de Educación, Universidades y Empleo under Grants CIAICO/2023-GVRTE/2023/4572860 and CIGE/2023/7; Ministerio de Ciencia e Innovación under Grant PID2021-128228NB-I00 and Fundación Mapfre under Grant “Modelización espacial e intra-anual de la mortalidad en España. Una herramienta automática para el cálculo de productos de vida.” 

Reviewers' comments:

Reviewer's Responses to Questions

**Comments to the Author**

1. Is the manuscript technically sound, and do the data support the conclusions?

Reviewer #1: Partly

Reviewer #2: Partly

2. Has the statistical analysis been performed appropriately and rigorously? 

Reviewer #1: Yes

Reviewer #2: No

3. Have the authors made all data underlying the findings in their manuscript fully available?

Reviewer #1: Yes

Reviewer #2: Yes

4. Is the manuscript presented in an intelligible fashion and written in standard English?

Reviewer #1: Yes

Reviewer #2: Yes

5. Review Comments to the Author

Reviewer #1: This is an interesting paper presenting a tool for building up a sub-annual life table - a different perspective from what is traditionally done in demography and public health sciences.

Even though the title and the introduction suggest that the objective of the paper will be on presenting the package qlifetable, I believe in some moments the authors took a different route and focus on the advantages of their approach of building up sub-annual life tables. In fact, this methodology was developed in other papers that they frequently refer to. I believe the authors can make it clear and concise throughout the paper that the main goal is to present the package itself and explore its functionalities and examples.

However, if selling the use of their sub-annual approach is also a goal, then I believe they should include some more evidence. First, I would state it clearly in the introduction that this is a goal of the paper as well. Second, as far as I know, demographers that follow either Preston et al (2001) or Wachter (2014) use the same Lexis-diagram (or should use at least) approach to compute mortality rates. So, if faced with the problem of calculating quarterly mortality rates, they usually calculate them on the same manner whenever the data is available in such a format. Therefore, in principle, this approach is not new. In fact, the innovation comes from the use of smoothing using seasonal-ageing indexes, which bring me to the third point. How much better are this approach compared to simply fitting a GAMS with death counts following a Poisson distribution smoothing over age and time? The authors do not compare their method/idea with other commonly used ones.

Further, given the amount of text that the authors devote for describing the method, few space is left for exploring the package’s functionalities, which they do very quick. For instance, they should spend more time explaining (and showing the structure) of the main data inputs. For example, do we need exact date of birth for each individual of the population? If so, this is rarely publicly available from censuses of most countries. What should we do in that case for using the package? If that’s a strong requirement, it should be clearly stated as a limitation.

I have some further minor comments. On the 5th paragraph of the introduction, the authors say that ‘knowing sub-annual risks does not contribute significantly to studying population trends and to developing long term projections and planning’. That is not true, sub-annual risks are key for understanding seasonal variations in mortality. Also, in the final part of section 3, the authors present the qx calculation formula – just make sure to state your assumptions, e.g., mention that you are assuming that those that died within that quarter die contribute for half a quarter person-year. Finally, state clearly the major contributions and limitations of the paper and the package in the end when compared with other standard ways of constructing life tables.

Reviewer #2: The manuscript presents an interesting approach to estimating life tables on a quarterly basis, taking into account aspects such as the seasonality of mortality data over the course of a year. Additionally, the authors introduce an R package for performing these estimations.

However, the manucripst needs to improve in several points. First, the structure of the manuscript is somewhat confusing. It is difficult to discern whether the primary aim is to introduce a new method or to present an R package. The division between explaining the method and then introducing the package creates some confusion. I would suggest the authors reconsider the organization, perhaps reducing the level of detail provided for the method (as it has been published elsewhere) and instead focusing on a more comprehensive description of the R package.

Furthermore, I have some concerns regarding the estimates of the annual and quarterly life tables, specifically in relation to the data used and its presentation in the Lexis diagram. The package takes inputs such as deaths, births, and migration data, which seems to complicate the traditional concept of life tables, typically based on population stationarity. Including migration data may confuse readers in this regard. This brings me to a second point: the data itself. The package seems to require true cohorts information, or at least paired birth and death data, which are available in very few countries globally. Consequently, this limits the package's applicability, especially in regions with limited data availability, such as middle- and low-income countries. As the manuscript primarily uses data from Spain, it would be beneficial to see the package tested on data from other regions to demonstrate broader applicability.

There is also a noticeable lack of comparison between these estimates and more conventional ones. Could the conventionally estimated quarterly tables (cross-sectional and period life table) yield results that differ significantly from the model presented? Lastly, the smoothing model using SAI is somewhat confusing in terms of its operation, which may be due to the structure adopted by the authors, dividing the explanation into two separate blocks.

6. PLOS authors have the option to publish the peer review history of their article (what does this mean?). If published, this will include your full peer review and any attached files.

Reviewer #1: No

Reviewer #2: No

---

## [Author Response · Author response to Decision Letter 0]

24 Oct 2024

Response to AE

qlifetable: An R package for constructing quarterly life tables

# PONE-D-24-36834

(Associate Editor)

Dear Associate Editor,

We wish to thank you for taking the time to review our paper and for considering that our paper is very interesting and relevant. The helpful suggestions and comments made by yourself and the two reviewers have without doubt significantly improved the paper. 

In order to make it easier to follow the revision and our answers, at the end of this note we have included your comments, encoded using Comment, and our responses, encoded using Response. 

Following your suggestions and comments and those made by the reviewers, we have made several changes and modifications to the paper. To easily track modifications, we have highlighted changes in the new version of the paper. Specifically, new additions have been highlighted in red, and alterations involving the repositioning of existing sentences in blue. Deletions have also been left in order to easily track the changes. In what follows we answer your queries and detail the changes introduced in the new version of the paper.

Comment 1. See reviewers’ comments. 

Response 1. We have considered all reviewers’ comments and addressed their concerns. Please see our responses to the reviewers.

Comment 2. Organization of the material should be revised.

Response 2. As stated in the responses to reviewers’ comments, we have clearly stated the twofold goal of our paper and better explained the organization of the paper. Please see the changes included in the paper and our responses to the first comment of both reviewers.

Comment 3. Provide more information on the functionalities of the package. 

Response 3. We have enlarged this part, providing more information on the functionalities of the package. Please see the changes introduced in Sections 5 and 6, including the new Figure 2.

Comment 4. Improve discussion on the limitations and applicability.

Response 4. We have improved the discussion on the limitations and applicability of the package. Please the changes included throughout the paper and those included in the last section.

Comment 5. More detail on how to organize data and other information.

Response 5. This has been addressed while providing more information about how to use the package, please see changes in Sections 5 and 6.

We hope you find the changes introduced adequate and the new version of the paper now suitable for publication in PLOS ONE.  

Response to reviewers

qlifetable: An R package for constructing quarterly life tables

# PONE-D-24-36834

(Reviewer #1)

Dear Reviewer #1,

We wish to thank you for taking the time to review our paper, for considering that the paper is interesting, and for all the detailed comments you have provided. The helpful suggestions and comments made by yourself, the Associate Editor and the other reviewer have without doubt significantly improved the paper. 

In order to make it easier to follow the revision and our answers, at the end of this note we have included your comments, encoded using Comment, and our responses, encoded using Response. 

Following your suggestions and comments and those made by the other reviewer and the editor, we have made several changes and modifications to the paper. To easily track modifications, we have highlighted changes in the new version of the paper. Specifically, new additions have been highlighted in red, and alterations involving the repositioning of existing sentences in blue. Deletions have also been left in order to easily track the changes. In what follows we answer your queries and detail the changes introduced in the new version of the paper.

Comment 1. This is an interesting paper presenting a tool for building up a sub-annual life table - a different perspective from what is traditionally done in demography and public health sciences. 

Even though the title and the introduction suggest that the objective of the paper will be on presenting the package qlifetable, I believe in some moments the authors took a different route and focus on the advantages of their approach of building up sub-annual life tables. In fact, this methodology was developed in other papers that they frequently refer to. I believe the authors can make it clear and concise throughout the paper that the main goal is to present the package itself and explore its functionalities and examples.

Response 1. Thank you for the comment. It has been very helpful. You are right that the aim of the paper was not clearly stated. To address this, we have clarified better the objective of the paper, which is twofold.

Our first goal is to document in detail how basic computations are performed in the package, specifically how time of exposures of risks are measured. Although this task does not entail special difficulties when one considers just one temporal dimension or when one works in an annual basis, it becomes particularly tricky when one considers two temporal dimensions and deals with sub-annual periods. Indeed, how to solve the lack of congruence (matching) between times measured working separately in the two dimensions when one considers the most straightforward and evident solution was an important challenge until we came up with a seemingly counterintuitive solution consisting in considering that the length of a year should not have exactly the same length for all the people. In our view, this is a new result that deserves to be communicated and clearly stated. Of course the package also keeps the option of making computations considering years of constant size. 

Sections 2 to 4 are dedicated to address this first goal. We devote Section 2 to detail, with the help of the Lexis scheme, how summary statistics, necessary to estimate quarterly life, tables can be derived from big microdata, Section 3 to briefly describe how summary statistics should be combined to produce quarterly tables, and Section 4 to document the issues related with the tricky issues related to the measurement of time. 

Our second goal is to describe all the functions in the package and to exemplify how they can be used to estimate SAIs, from which quarterly tables can be derived from annual tables. We devote Sections 5 and 6 to this task. Section 5 introduces all the functions and Section 6 illustrates, though a complete example, how to use them.

The focus of the paper is not detailing the advantages of our particular approach for building up sub-annual life tables, which have been stated elsewhere. Although as a motivation and to highlight the advantages of having a tool as the one is presented in the paper, we make in the introduction some comments about the advantages of having quarterly life tables. To avoid the potential misleading effects of our comments over the focus of the paper and to address your concerns, we have made some amendments in the paper eliminating sentences about the advantages of using our approach. 

Please see the changes introduced in the abstract and the introduction to address all these issues.

Comment 2. However, if selling the use of their sub-annual approach is also a goal, then I believe they should include some more evidence. First, I would state it clearly in the introduction that this is a goal of the paper as well. Second, as far as I know, demographers that follow either Preston et al (2001) or Wachter (2014) use the same Lexis-diagram (or should use at least) approach to compute mortality rates. So, if faced with the problem of calculating quarterly mortality rates, they usually calculate them on the same manner whenever the data is available in such a format. Therefore, in principle, this approach is not new. In fact, the innovation comes from the use of smoothing using seasonal-ageing indexes, which bring me to the third point. How much better are this approach compared to simply fitting a GAMS with death counts following a Poisson distribution smoothing over age and time? The authors do not compare their method/idea with other commonly used ones.

Response 2. Thank you for the comment. As stated in the previous response, this is not the goal of the paper, and we believe we have properly addressed it in the new version. Regarding your comment, to the best of our knowledge, our approach differs from those currently followed in demography to estimate sub-annual risks. In our understanding, while existing methods may consider the age dimension when addressing sub-annual risks, they often omit the seasonal dimension or only measure time-distances between events using the calendar dimension. In contrast, we explicitly consider both the calendar and age dimensions simultaneously, leading to the construction of four quarterly tables instead of a single one. We are very grateful for the suggestion, which deserves further exploration in future research.

Comment 3. Further, given the amount of text that the authors devote for describing the method, few space is left for exploring the package’s functionalities, which they do very quick. For instance, they should spend more time explaining (and showing the structure) of the main data inputs. For example, do we need exact date of birth for each individual of the population? If so, this is rarely publicly available from censuses of most countries. What should we do in that case focusing the package? If that’s a strong requirement, it should be clearly stated as a limitation.

Response 3. Thank you very much for the recommendations. This comment has three issues to be addressed. First, to address the misunderstanding related to the space devoted to describing the method, we have clarified the twofold goal of the paper. Section 3, which is the shorter in the paper, is the one when it is quickly summarized the methodological details of the approach, which we consider necessary to understand the main outputs of the functions. The rest of the paper is devoted to either explain how computations are made in the package (Sections 2 to 4) or explain how to use the package (Sections 5 and 6). We are confident that with the changes introduced this is more clearly stated in the new version of the paper. Second, to address the sometimes brief and quick explanations about the use and requirements of the package, we have enlarged this explaining how to use it in more detail, including the structures of the data inputs and outputs. We refer to the changes included in Sections 5 and 6 for this. Third, we have explained the different alternatives the package offers to estimate quarterly death rates and derive SAIs depending on the level of detailed available information and better stated its limitations in the conclusions. Please see the changes introduced in Sections 5 to 7, including the new Figure 2 in Section 5.

Comment 4. I have some further minor comments. On the 5th paragraph of the introduction, the authors say that ‘knowing sub-annual risks does not contribute significantly to studying population trends and to developing long term projections and planning’. That is not true, sub-annual risks are key for understanding seasonal variations in mortality. Also, in the final part of section 3, the authors present the qx calculation formula – just make sure to state your assumptions, e.g., mention that you are assuming that those that died within that quarter die contribute for half a quarter person-year. Finally, state clearly the major contributions and limitations of the paper and the package in the end when compared with other standard ways of constructing life tables.

Response 4. Thanks for the comment. There are again three issues to address here. First, you are right that the sentence could be misleading. While we agree that sub-annual risks are crucial for understanding seasonal variations in mortality, we need to clarify that our comment was meant to indicate that quarterly tables are not necessary for making long-term population projections, as similar population figures can be obtained using annual tables. We have removed that sentence to avoid confusion. Second, you're correct that the formula we present to derive qx from mx assumes that those who died within the quarter contribute half a quarter person-year, an assumption not made explicit when deriving mx. We have clarified this in the revised version of the paper. Third, we have enhanced the discussion of the major contributions and limitations of the approach programmed in the package.

We hope you find the changes introduced adequate and the new version of the paper now suitable for publication in PLOS ONE.

Response to reviewers

qlifetable: An R package for constructing quarterly life tables

# PONE-D-24-36834

(Reviewer #2)

Dear Reviewer #2,

We wish to thank you for taking the time to review our paper, for sharing with us that the package offers an interesting approach to estimating life tables on a quarterly basis, and for all the detailed comments you have provided. The helpful suggestions and comments made by yourself, the Associate Editor and the other reviewer have without doubt significantly improved the paper. 

In order to make it easier to follow the revision and our answers, at the end of this note we have included your comments, encoded using Comment, and our responses, encoded using Response. 

Following your suggestions and comments and those made by the other reviewer and the editor, we have made several changes and modifications to the paper. To easily track modifications, we have highlighted changes in the new version of the paper. Specifically, new additions have been highlighted in red, and alterations involving the repositioning of existing sentences in blue. Deletions have also been left in order to easily track the changes. In what follows we answer your queries and detail the changes introduced in the new version of the paper.

Comment 1. The manuscript presents an interesting approach to estimating life tables on a quarterly basis, taking into account aspects such as the seasonality of mortality data over the course of a year. Additionally, the authors introduce an R package for performing these estimations.

However, the manuscript needs to improve in several points. First, the structure of the manuscript is somewhat confusing. It is difficult to discern whether the primary aim is to introduce a new method or to present an R package. The division between explaining the method and then introducing the package creates some confusion. I would suggest the authors reconsider the organization, perhaps reducing the level of detail provided for the method (as it has been published elsewhere) and instead focusing on a more comprehensive description of the R package.

Response 1. Thank you for the comment. It has been very helpful. You are right that the aim of the paper was not clearly stated in the previous version, which led to some confusion. The goal of the paper is not to introduce a new method. We only dedicate Section 3 (the shortest section of the paper) to briefly explain the mathematics of the method, which we consider relevant for understanding the outputs of the different functions. To address this confusion, we have clarified the twofold goal of the paper. Please refer to the changes in the abstract and introduction. On the one hand, we have clarified that one of our aims is to document in detail how the basic summary statistics are computed, particularly how exposure times are measured in the package, as this is a complex issue when considering two temporal dimensions and sub-annual periods. On the other hand, our second aim is to describe the inputs required and the outputs generated by the package's functions, including examples of how they can be used to estimate SAIs and quarterly life tables. Please also refer to our response to Comment 1 of Reviewer #1, as well as the changes made throughout the paper.

Comment 2. Furthermore, I have some concerns regarding the estimates of the annual and quarterly life tables, specifically in relation to the data used and its presentation in the Lexis diagram. The package takes inputs such as deaths, births, and migration data, which seems to complicate the traditional concept of life tables, typically based on population stationarity. Including migration data may confuse readers in this regard

---

## [Decision Letter · Decision Letter 1]

4 Dec 2024

qlifetable: An R package for constructing quarterly life tables

PONE-D-24-36834R1

Dear Dr. Pavía,

We’re pleased to inform you that your manuscript has been judged scientifically suitable for publication and will be formally accepted for publication once it meets all outstanding technical requirements.

Kind regards,

Bernardo Lanza Queiroz, Ph.D

Academic Editor

PLOS ONE

Additional Editor Comments (optional):

Reviewers' comments:

Reviewer's Responses to Questions

**Comments to the Author**

1. If the authors have adequately addressed your comments raised in a previous round of review and you feel that this manuscript is now acceptable for publication, you may indicate that here to bypass the “Comments to the Author” section, enter your conflict of interest statement in the “Confidential to Editor” section, and submit your "Accept" recommendation.

Reviewer #1: All comments have been addressed

Reviewer #2: (No Response)

2. Is the manuscript technically sound, and do the data support the conclusions?

Reviewer #1: Yes

Reviewer #2: Yes

3. Has the statistical analysis been performed appropriately and rigorously? 

Reviewer #1: Yes

Reviewer #2: Yes

4. Have the authors made all data underlying the findings in their manuscript fully available?

Reviewer #1: Yes

Reviewer #2: Yes

5. Is the manuscript presented in an intelligible fashion and written in standard English?

Reviewer #1: Yes

Reviewer #2: Yes

6. Review Comments to the Author

Reviewer #1: (No Response)

Reviewer #2: (No Response)

7. PLOS authors have the option to publish the peer review history of their article (what does this mean?). If published, this will include your full peer review and any attached files.

Reviewer #1: No

Reviewer #2: No

---

## [Editor Report · Acceptance letter]

13 Dec 2024

PONE-D-24-36834R1 

PLOS ONE

Dear Dr. Pavía, 

I'm pleased to inform you that your manuscript has been deemed suitable for publication in PLOS ONE. Congratulations! Your manuscript is now being handed over to our production team.

Kind regards, 

on behalf of

Dr. Bernardo Lanza Queiroz 

Academic Editor

PLOS ONE